# Prior Metadata-Driven RAW Reconstruction: Eliminating the Need for Per-Image Metadata

## ABSTRACT

While RAW images are efficient for image editing and perception tasks, their large size can strain camera storage and bandwidth. Reconstruction methods of RAW images from sRGB data typically require additional metadata from the RAW image, which increases camera processing computations. To address this problem, we propose using Prior Meta as a reference to reconstruct the RAW data instead of relying on per-image metadata. Prior metadata is extracted offline from reference RAW images, which are usually part of the training dataset and have similar scenes and light conditions as the target image. With this prior metadata, the camera does not need to provide any extra processing other than the sRGB images, and our model can autonomously find the desired prior information. To achieve this, we design a three-step pipeline. First, we build a pixel searching network that can find the most similar pixels in the reference RAW images as prior information. Then, in the second step, we compress the large-scale reference images to about 0.02% of their original size to reduce the searching cost. Finally, in the last step, we develop a neural network reconstructor to reconstruct the high-fidelity RAW images. Our model achieves comparable, and even better, performance than RAW reconstruction methods based on metadata.

## CCS CONCEPTS

• **Art and Culture → Art and Culture**.

## KEYWORDS

RAW Image, sRGB, Metadata, Reconstruction, Prior Metadata

## 1 INTRODUCTION

RAW images are unprocessed radiance captured by the camera, with a larger bit width of usually 10-16 bits, compared to the 8-bit sRGB data of compressed images. This gives RAW data wider light tolerance and retains more detailed information for dark and high-dynamic scenes [21, 22, 38]. Additionally, RAW images have the advantage of maintaining their original linear status, as they are not processed by non-linear modules in the ISP [6, 36]. This makes them more suitable for deep learning models to understand the original information distribution. Numerous studies have demonstrated the great potential of RAW images in computer vision tasks such as super-resolution [36, 39, 41], denoising [2, 11, 20, 21], and object

Permission to make digital or hard copies of all or part of this work for personal or classroom use is granted without fee provided that copies are not made or distributed for profit or commercial advantage and that copies bear this notice and the full citation on the first page. Copyrights for components of this work owned by others than the author(s) must be honored. Abstracting with credit is permitted. To copy otherwise, or republish, to post on servers or to redistribute to lists, requires prior specific permission and/or a fee. Request permissions from permissions@acm.org.
*ACM MM, 2024, Melbourne, Australia*
© 2024 Copyright held by the owner/author(s). Publication rights licensed to ACM.
ACM ISBN 978-x-xxxx-xxxx-x/YY/MM
https://doi.org/10.1145/nnnnnnn.nnnnnnn

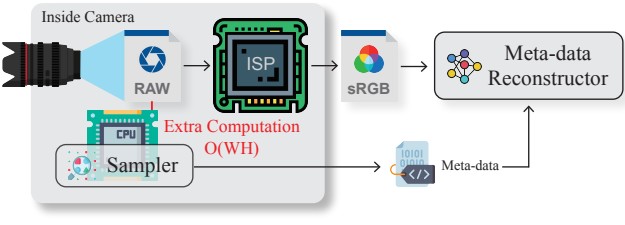

(a) Meta-data Based Methods

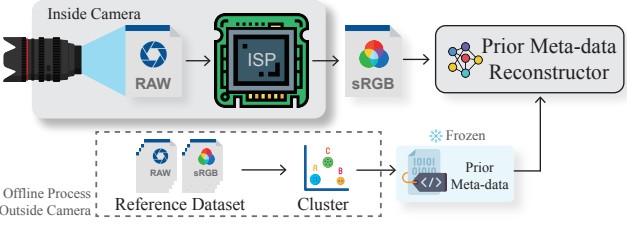

(b) Our Method

**Figure 1: Comparison with Previous RAW Reconstruction Methods. (a) Methods based on metadata introduce additional computational burden to the mobile camera. (b) Our method does not require the camera to provide additional information. Instead, it searches for similar pixels from the reference RAW images as a reference.**

detection [13, 27, 38]. However, the high cost of collecting RAW data limits its application in many scenarios. On the one hand, the detailed information preserved in RAW images results in a much larger file size than their corresponding sRGB images, which can impose significant storage and bandwidth burdens on cameras when capturing large amounts of data. On the other hand, most labeled data for computer vision tasks are based on sRGB data without corresponding RAW data, which limits the applicability of RAW data in many perception tasks.

To address these limitations, several methods have been proposed to reconstruct the RAW images based on their corresponding sRGB images. One approach involves using a deep neural network to de-render the RAW images, using only the sRGB data as input [9, 37, 42]. These methods are convenient as they don't require any additional inputs besides the sRGB information. However, they may suffer from relatively lower performance since the ISP processing removes detailed information from the original RAW data. It is challenging for deep neural networks to perfectly recover these dropped details without proper reference. Another approach is to use extra metadata as a reference to improve the accuracy [17, 24, 28, 33]. These methods employ a sampler in the camera to sample some RAW pixels into the metadata. When de-rendering the RAW data, another reconstructor is used, which takes the sRGB images and corresponding metadata as input to produce

 

the original RAW images, as shown in Fig. 1(a). These methods achieve good performance in terms of the quality of the predicted RAW images. However, the sampler running in the camera device adds an additional computational burden, resulting in $O(WH)$ computation and space complexity, where $W$ and $H$ represent the width and height of the RAW image. Also, these methods cannot be used to recover existing sRGB images without metadata. This limitation restricts the use of existing sRGB datasets for RAW-based computer vision tasks.

In contrast, we propose to search for similar pixels in existing RAW images as a reference, instead of relying on online metadata sampling when capturing new photos. Therefore, our method does not introduce any additional computation and storage on the camera device. In other words, the complexity of our method on the camera is 0. While each photo is unique and distinct from all others, at the pixel level, we can identify similar pixels that were captured under similar lighting conditions and depict similar objects. Although these pixels may not be an exact match for the target pixel, they can provide valuable prior information for the deep neural network. Therefore, we utilize these pixels as references and refer to them as *Prior Metadata*, as shown in Fig. 1(b).

To achieve this goal, we first train a pixel searching model using contrast learning. This model takes sRGB data as input and generates an embedding vector for each pixel in the image. We then minimize the vector distance if the corresponding RAW pixels are similar and increase the distance if the RAW pixels are different. Using this model, we can identify the most similar RAW pixels in the reference dataset and assemble prior RAW images using these pixels. Based on the prior RAW images, we build a reconstructor based on the prior RAW images and the corresponding sRGB images. However, when the reference dataset is very large, the searching procedure becomes computationally and memory intensive. On the other hand, if the reference dataset is too small, it may not contain enough pixel variety to build the prior RAW images. To overcome this challenge, we design a clustering and grouping algorithm that selects several representative pixels from the entire reference dataset to replace the actual reference dataset. Based on our analysis and experiments, we have found that sampling around 5,000 pixels can effectively represent the entire dataset and achieve similar performance. This accounts for only about 0.02% of the total pixels in the reference dataset. Experiments demonstrate that our method achieves impressive RAW construction accuracy. Even without the online metadata sampled from the target RAW images, our method can effectively construct the RAW images compared to state-of-the-art online metadata-based methods. In summary, our contribution can be summarized in four aspects:

- We propose a new pipeline for extracting prior metadata from the existing RAW images and employing it as a reference for RAW reconstruction.
- We propose a contrastive training approach to train the Pixel Searching Network. This network enables the search for the most similar RAW pixels in the reference dataset, using features extracted from sRGB images.
- We propose a prior metadata clustering and grouping algorithm that can identify the most representative pixels for the reference dataset, thus reducing the search cost.

- We propose a RAW Reconstructor with a Prior Metadata Fusion Module. This approach efficiently utilizes reliable information encoded in the prior metadata while discarding unreliable information. The model achieves excellent performance in RAW reconstruction, outperforming metadata-based methods that require per-image metadata extracted from the target RAW.

## 2 RELATED WORKS

### 2.1 RAW Image Application

The application of RAW data to low-level tasks has attracted more and more attention in computer vision. For example, relying on the linear relationship between RAW data and scene radiation, image denoising and image super-resolution tasks benefit from it. The work in [2] designs a RAW domain denoising network using synthetic noise data by adding shot noise and read noise, which gets significant improvements in denoising capabilities compared to state-of-the-art RGB-based denoising models. The work in [35] proposes a highly accurate noise model based on the characteristics of CMOS photosensors to address the extreme low-light RAW denoising problem. In [36], Xing et al. present an end-to-end joint learning framework for demosaicing, denoising, and super-resolution, which enables satisfactory results to be achieved on multiple visual tasks based on RAW data. In addition, benefiting from a bigger dynamic range, RAW data also perform better in the low-light enhancement task. It is interesting to note that Chen et al. [6] build a dataset with low-light images and corresponding long-exposure high-quality images, and the promising results inspire future work. After this, the work in [10] and [16] improve the accuracy and speed of the model, respectively.

Except for the above-mentioned low-level tasks, the advantages of RAW data in high-level tasks have also been discovered, such as object detection and segmentation. On the one hand, since ISP processing is only for human eye viewing, it is a practical problem to remove unnecessary steps of ISP to obtain faster processing speed, which is crucial for some real-time systems that require low time consumption and fast response, such as autonomous driving. As for related works, [23, 30, 40] are dedicated to configuring ISP as learnable procedures. Through the end-to-end optimization process of the neural network, we are able to skillfully mine ISP operations that are beneficial to downstream tasks and configure their parameters reasonably. On the other hand, a larger bit width can provide more information for perception tasks in dark scenes and highly dynamic scenes. Works [38] and [7] respectively prove that RAW data with higher bits can achieve better results in object detection and instance segmentation tasks under dark light conditions. In work [27], the researchers propose a neural auto-exposure system that jointly trains an object detector and an image signal processing pipeline. Experiments on the vehicle detection data set prove that reasonable processing of RAW data can greatly improve the detection accuracy of the model in HDR scenes. Even though the benefits of the RAW format in these tasks have been discovered, further exploration is needed.

## 2.2 RAW Reconstruction Methods

Due to the difficulty in obtaining RAW data, the convenient use of sRGB images to reverse-generate RAW images has gradually become a hot topic. According to whether they rely on metadata information, popular sRGB2RAW methods are classified into two categories.

**Blind RAW reconstruction.** Image signal processing enhances the captured image to make it more visually appealing to humans. This is achieved through various steps, such as white balance, demosaicing, and gamma correction, many of which can be explicitly modeled. As a result, some methods for reconstructing RAW images [2, 9] aim to reverse certain key steps of modern ISPs in order to obtain the unprocessed RAW image. For example, [9] models six key stages of the ISP, including demosaic, lens shading correction, white balance, color correction, gamma correction, and tone mapping. For camera-specific modules (such as color correction), this method represents them by constructing a learnable dictionary. While these methods offer strong interpretability, they often lack generalization ability due to the specific nature of the camera ISP. To overcome this limitation, other approaches [37, 42] propose using the concept of cycle constraints to simultaneously learn the bidirectional process of RAW-to-sRGB and sRGB-to-RAW conversion. For instance, [37] proposes using the Normalizing Flow model to learn an invertible ISP network. Due to the inherent reversibility of the Normalizing Flow model, we can conveniently perform RAW-to-sRGB conversion and its inverse process. Unfortunately, due to the inevitable information loss in ISP, blind RAW reconstruction methods struggle to achieve satisfactory results.

**RAW reconstruction with metadata.** Metadata-based methods leverage additional information to assist in the reconstruction process, such as ISP parameters [25], sampled RAW pixels [17, 24, 28], and latent features [33]. Early methods [25] store key parameters in the ISP forward rendering process, such as color correction matrices, auto white balance parameters, etc., but the obvious disadvantage is that some key operations cannot be quantitatively stored, such as tone mapping. The most commonly used sampling-based methods employ strategies to save a subset of pixels from the original resolution RAW image. For instance, methods in [17, 28] utilize uniformly sampled pixels as metadata and then employ interpolation algorithms or neural networks to recover the remaining pixels. The work in [28] learns the content-related pixels using a sampler network, and then performs an efficient recovery process through a U-Net network. These metadata provide reliable auxiliary information for RAW image recovery, but generating corresponding metadata for each image on the camera side can be time-consuming and labor-intensive.

In contrast to the aforementioned methods, our model directly extracts representative pixels from existing RAW images as prior information, which serves as guidance and eliminates the need for an additional metadata generator.

## 3 METHOD

### 3.1 Pipeline

To achieve prior metadata-based RAW reconstruction, we propose a new pipeline shown in Fig. 1(b). The pipeline consists of three main steps. The first step is training a Pixel Searching Network,

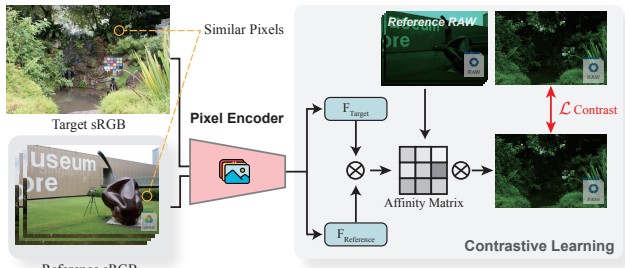

**Figure 2: Illustration of the Pixel Searching Pipeline. The first step in our pipeline is to train a pixel searching model. This model is used to search for the most relevant pixels from the reference dataset for the target pixels.**

which converts sRGB pixels into a RAW-aware embedding. This embedding allows us to search the entire reference dataset for similar pixels to the target pixel. These similar pixels can then be used as prior information for RAW image reconstruction. This approach is based on the observation that archived RAW images captured by the same type of devices (same sensor) can provide valuable information for the RAW reconstruction process. At the pixel level, previous RAW pixels captured in similar scenes with similar lighting conditions can serve as high-quality references for recovering the current pixel. The design and training of the Pixel Searching Encoder will be discussed in detail in § 3.2.

To achieve better generalization, the reference dataset should contain a variety of scenes with different lighting conditions and objects. However, this would require a large-scale reference dataset, causing a significant computational burden when searching for similar pixels. To address this issue, we propose the second step in our pipeline: clustering and grouping the pixels into a few of the most representative ones. Despite the large size of the entire reference dataset, many pixels exhibit similarity. Therefore, we can evaluate the similarity between the pixels in the reference dataset and select a few highly representative pixels to represent the entire dataset. To accomplish this, we design a clustering and grouping algorithm, which will be introduced in detail in § 3.3.

While similar pixels in the reference dataset can provide valuable prior information, the assembled RAW image created from these pixels may not perfectly match the target RAW. Therefore, a deep neural network is required to address these differences. To fully utilize the prior information from the prior metadata and the target sRGB image, we introduce a Prior Metadata Fusion Module. This module incorporates all useful information for RAW reconstruction. We then use an encoder-decoder network to reconstruct the target RAW based on the fused inputs. This procedure will be demonstrated in Section § 3.4.

### 3.2 Pixel Searching Encoder

The transformation from RAW to sRGB, denoted as $sRGB = f(RAW)$, is an information-loss procedure. As a result, the inverse transformation, $RAW = f^{-1}(sRGB)$, is not a one-to-one mapping. This

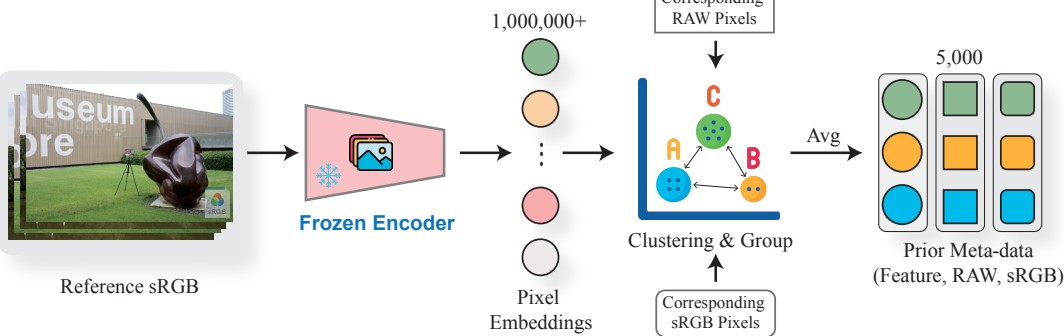

**Figure 3: Illustration of the Prior Information Compression Method. To optimize the searching computation, the next step in our pipeline involves clustering and grouping the reference dataset into compressed prior metadata. This compressed metadata is significantly smaller, comprising only about 0.02% of the original size.**

means that the same sRGB pixels can represent different RAW values depending on the context. This feature makes it challenging to identify similar RAW pixels in the reference images based on the sRGB values of the target images. Instead of searching for similar pixels based on the similarity of sRGB values, we develop a deep neural network that learns to find similar RAW pixels using an sRGB image as input. This network converts the surrounding sRGB context of the pixel into a high-dimensional vector and evaluates the distance between the vectors to identify the most similar RAW pixels. When evaluating the distance between pixels, the Pixel Searching Encoder takes into account not only the similarity between their sRGB values but also the lighting conditions and scenes depicted in the context patch. This approach enhances the efficiency of finding similar RAW values.

The Pixel Searching Encoder adopts a ResNet [12]-like structure as a feature extractor with two major modifications. On the one hand, the similarity among RAW pixels is considered a low-level feature, so we decrease the parameter number of the network to prevent overfitting. On the other hand, the down-sample ratio of the encoder will affect the resolution of the feature map. To maintain a higher resolution of the pixel encoding, we adjust the total down-sampling ratio of the encoder to 2. This means that the output resolution of the encoding feature map is half the size of the original sRGB image.

To effectively train the Pixel Searching Encoder, we propose a contrastive training method as illustrated in Fig. 2(a). The main idea is to bring sRGB features closer together if their RAW values are similar and push them apart if they are not. To achieve this, we first randomly divide the original training dataset into two subdatasets: a target dataset and a reference dataset. The images in the reference dataset will serve as references and will be used to search for similar RAW pixels for the target image. During the training process, we randomly select batches from the target dataset and separate batches from the reference set as references. We input the target and reference images into the Pixel Searching Encoder to extract feature maps:

$$f_t = \text{Encoder}(\text{sRGB}_t)$$
$$f_r = \text{Encoder}(\text{sRGB}_r). \qquad (1)$$

We then multiply the two feature maps to generate an affinity matrix between the target and reference images:

$$\mathcal{A} = f_t \otimes f_r. \qquad (2)$$

Finally, we apply a softmax function to the affinity matrix in the reference dimension and multiply it with the reference RAW pixels to produce an assembled RAW image:

$$\mathcal{F}\text{RAW}_t = \text{Softmax}(\mathcal{A}) \otimes \text{RAW}_r. \qquad (3)$$

The assembled RAW image $\mathcal{F}\text{RAW}_t$ is utilized to calculate the L2 loss with the actual target RAW images. During this process, the values of the reference pixels remain fixed, and only the affinity matrix can be optimized using the gradient descent algorithm. Consequently, the Pixel Searching Encoder will learn to maximize the inner product between feature embeddings for similar RAW pixels and minimize it for dissimilar ones.

## 3.3 Prior Information Compression

The computation of the affinity matrix has a space and time complexity of $O(MLN)$, where $M$ and $N$ represent the number of pixels in each target and reference image, and $L$ is the size of the reference dataset. During the inference phase, this calculation becomes computationally intensive if the reference dataset size $L$ is very large. However, it is important to have a diverse reference dataset that includes various conditions where the target images may be captured. Only with this diverse reference dataset, the model can efficiently select the most similar pixel for the target image.

In addition to using reference images to assemble the target RAW images, we can also assemble the reference dataset itself by incorporating specific pixels from the reference dataset. This approach allows us to select a subset of the most representative pixels from the dataset to represent the entire reference dataset. To achieve this, as shown in Fig. 2(b), we propose a clustering and grouping algorithm that effectively selects and generates the representative pixels for the reference dataset.

We first manually select 100 representative images from the training dataset as a reference dataset. These images are then converted into feature maps using the Pixel Searching Encoder trained with our contrastive learning method. Each image in the reference

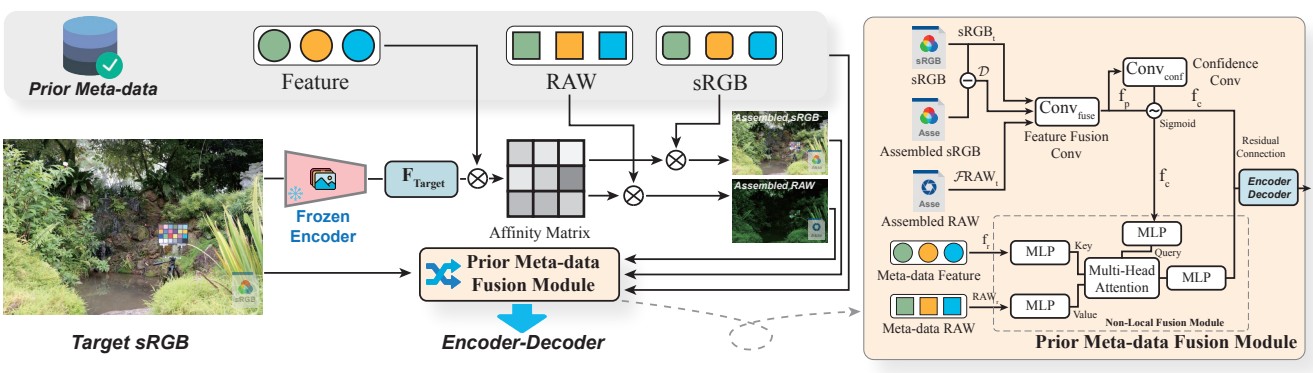

**Figure 4: Illustration of the Reconstructor. The prior metadata is generated offline and remains fixed. Our reconstructor only needs the target sRGB during inference. To effectively use the prior metadata, we propose the Prior Metadata Fusion module, which combines features from the sRGB image and the prior metadata.**

dataset is cropped and resized into a resolution of $1024 \times 1024$, resulting in output feature maps with a resolution of $512 \times 512$. The reference dataset contains a total of $26, 214, 400$ pixels. We then use the K-Means algorithm to cluster the feature embeddings into several different groups. Since the reference dataset has a large number of sampling points, the clustering procedure requires significant computation resources. To address this issue, we utilize the faiss algorithm [14], which is a library for efficient similarity search and clustering of dense vectors. With this algorithm, we can perform the clustering procedure on a single computer within a few minutes, making our methods more easily applicable. In our experiment, we use the inner product as the clustering distance measurement and randomly select the initial centroid of each cluster. We cluster a total of $5, 000$ groups in our experiment and select the $200$ nearest features in the reference dataset for each group. Finally, we average the $200$ features and their corresponding sRGB and RAW pixel values to produce the final prior metadata. Compared to the original $26, 214, 400$ pixels, the $5, 000$ pixels in the prior metadata represent only about $0.02\%$ of the total. However, in our experiment, the $5, 000$ pixels in the prior metadata achieve virtually the same performance as the entire reference dataset, which will be introduced detailedly in § 4.

## 3.4 Prior Metadata Based Reconstructor

By utilizing the grouped prior metadata and the Pixel Searching Encoder, we can assemble a target RAW image through similarity searching. However, it is important to understand that the assembled RAW image cannot be directly used as reconstructed RAW data because of slight discrepancies with the actual RAW image at each pixel. In some cases, there may be pixels in unique scenarios that cannot find exact matches. Additionally, fine details like edges may not be accurately represented by the reference pixels. To overcome this limitation, we have implemented a deep learning network to address and rectify these disparities.

As illustrated in Fig. 4, the reconstructor we propose comprises two primary modules: a Prior Metadata Fusion Module and a RAW Encoder Decoder Module. The initial stage in effectively reconstructing the desired RAW image is to efficiently merge all available

prior data. To accomplish this, we have developed the module to integrate the information encoded in the prior metadata and the target sRGB image.

**Prior Metadata Fusion Module:** Fig. 4 provides an overview of this module. To start, we transform the target sRGB image into a feature map for similarity searching using the well-trained Pixel Searching Encoder. Then, using the affinity matrix $\mathcal{A}$ that captures the relationship between the target features and the prior metadata features, we produce an assembled RAW image and its corresponding sRGB image:

$$\mathcal{F}sRGB_t = \text{Softmax}(\mathcal{A}) \otimes sRGB_r$$
$$\mathcal{F}RAW_t = \text{Softmax}(\mathcal{A}) \otimes RAW_r. \quad (4)$$

The assembled sRGB image can provide valuable information for assessing the accuracy of each pixel in the assembly. When the searched sRGB pixel has little difference from the target pixels, there is a high probability that the searched RAW pixel is similar to the target RAW pixel. However, due to the non-injective nature of this theory, it may not always be correct. Nonetheless, it can help filter out most of the unreliable pixels. Using this information, we calculate the distance $\mathcal{D}$ between the assembled sRGB image and the target sRGB image:

$$\mathcal{D} = abs(\mathcal{F}sRGB_t - sRGB_t). \quad (5)$$

Then, we combine this distance information with the target sRGB image and the assembled RAW image. We then send them into a convolutional network for prior-aware feature extraction:

$$f_p = \text{Conv}_{\text{fuse}}([\mathcal{D}, sRGB_t, \mathcal{F}RAW_t]). \quad (6)$$

To eliminate the unreliable pixels in this feature map, we utilize an additional convolutional network to generate spatial self-attention:

$$f_c = f_p \cdot \text{Sigmoid}(\text{Conv}_{\text{conf}}(f_p)). \quad (7)$$

In addition to providing detailed prior information to individual pixels in the generated target RAW image, the prior metadata can also serve as a global reference for reconstructing the target RAW image. To achieve this, we need the model to learn how to extract relevant information from the prior metadata. Therefore, we employ a multi-head attention mechanism that considers the feature $f_c$ as

the query, the prior feature $f_r$ as the key, and the prior RAW pixels $\text{RAW}_r$ as the value. During the training phase, this attention module is optimized in an end-to-end manner and effectively aggregates the desired global reference for each pixel in the feature map. Finally, the RAW reconstruction feature map $f_o$ is obtained using a feed-forward network (FFN):

$$f_o = FFN(MHA(f_c, f_r, \text{RAW}_r)). \qquad (8)$$

Finally, we use the Encoder-Decoder Module to reconstruct the final RAW image.

**RAW Encoder-Decoder Module:** We utilize an Encoder-Decoder structure, as described in [24], to accomplish the RAW reconstruction. This module takes the RAW reconstruction feature map as input and progressively reduces the features to a $\frac{1}{16}$ resolution, allowing for the encoding of information with a large receptive field. Following the encoding procedure, the decoder gradually up-samples the feature map and combines it with the corresponding feature maps from the encoder, in order to recover the detailed information that was lost during the down-sampling procedure. In contrast to previous structures, we additionally fuse the upsampled feature map with the RAW reconstruction feature map, as it encodes important detailed information from the assembled RAW image. To achieve this, we first interpolate the RAW reconstruction feature map to the same size as the upsampled feature map, and then employ a convolution operation to fuse the two feature maps. Finally, we utilize a head network to decode the feature map into 3-channel RAW images and limit the value range to 0-1 using a sigmoid function.

## 4 EXPERIMENTS

### 4.1 Implementation Detail

**Dataset** To evaluate the performance of our method for the RAW reconstruction task, we chose the representative NUS dataset [8] as a benchmark. This dataset consists of photos captured by three different cameras: Samsung NX2000, Olympus E-PL6, and Sony SLT-A57, with 202, 208, and 268 RAW images, respectively. Following the approach of Nam et al., all the RAW images were processed using the demosaic procedure and bilinear interpolation to obtain the 3-channel RAW-RGB image with the original resolution. To obtain the corresponding sRGB images, we used a software ISP emulator [15] to convert the RAW images into the sRGB format.

**Loss** In the training phase, the reconstruction network uses the sRGB images as input to assemble the prior RAW and prior sRGB images. This is done by taking the sRGB inputs and the prior metadata generated by the clustering and grouping algorithm. The model then produces the RAW prediction based on the sRGB images and the prior information. To determine the loss function, we utilize $L_2$ loss and $SSIM$ [34] loss as:

$$L = \alpha L_2(y, Y) + \beta SSIM(y, Y), \qquad (9)$$

where $y$ and $Y$ are the prediction and ground truth, respectively. $\alpha$ and $\beta$ are the balance ratio.

**Metrics** In this paper, we adopt PSNR (Peak Signal-to-Noise Ratio) and SSIM (Structural Similarity Index Measure) [34] as our main evaluation metrics, following precedents in the field. PSNR

---

**Algorithm 1** Contrastive Training Algorithm

1: **procedure** CONTRASTIVETRAINING($data_t$, $Encoder$)
2:     **for** each ($sRGB_t$, $sRGB_r$, $RAW_t$, $RAW_r$) in $data_t$ **do**
3:         $f_t \leftarrow Encoder(sRGB_t)$
4:         $f_r \leftarrow Encoder(sRGB_r)$
5:         $\mathcal{A} \leftarrow f_t \times f_r$
6:         $\mathcal{A}_s \leftarrow Softmax(\mathcal{A})$
7:         $\mathcal{F}\text{RAW}_t \leftarrow \mathcal{A}_s \times RAW_r$
8:         $loss \leftarrow Loss(\mathcal{F}\text{RAW}_t, RAW_t)$
9:         $Encoder.update\_weights(loss)$
10:     **end for**
11: **end procedure**

---

**Algorithm 2** Prior Information Compression Algorithm

1: **procedure** PRIORINFORMATIONCOMPRESSION($data_r$, $Encoder$)
2:     $f_rAll \leftarrow [Encoder(sRGB_r)$ for $sRGB_r$ in $rgb\_data_r]$
3:     $RAW_rAll \leftarrow RAW_r$ for $RAW_r$ in $raw\_data_r]$
4:     $sRGB_rAll \leftarrow sRGB_r$ for $sRGB_r$ in $rgb\_data_r]$
5:     $clusters \leftarrow KMeansClustering(f_rList)$
6:     **for** each $cluster$ in $clusters$ **do**
7:         $near\_idx \leftarrow selectNearest(cluster, 200)$
8:         $RAW_{avg} \leftarrow mean(near\_idx, RAW_rAll)$
9:         $sRGB_{avg} \leftarrow mean(near\_idx, sRGB_rAll)$
10:         $f_{avg} \leftarrow mean(near\_idx, f_rAll)$
11:         $store(RAW_{avg}, sRGB_{avg}, f_{avg}, prior)$
12:     **end for**
13: **end procedure**

---

**Algorithm 3** Prior Metadata Based RAW Reconstructing Algorithm

1: **procedure** RECONSTRUCTOR($sRGB_t$, $prior$, $Encoder$, $Decoder$)
2:     $f_t \leftarrow Encoder(sRGB_t)$
3:     $\mathcal{A}_s \leftarrow affinityMatrix(f_t, prior)$
4:     $\mathcal{F}sRGB_t, \mathcal{F}RAW_t \leftarrow assemble(\mathcal{A}_s, prior)$
5:     $diff \leftarrow abs(\mathcal{F}sRGB_t - sRGB_t)$
6:     $f_p \leftarrow Conv_{fuse}([diff, sRGB_t, \mathcal{F}RAW_t])$
7:     $f_c \leftarrow f_p \cdot Sigmoid(Conv_{conf}(f_p))$
8:     $f_o \leftarrow FFN(MHA(f_c, prior))$
9:     $Output \leftarrow UNet(f_o, sRGB_t)$
10: **end procedure**

---

measures the ratio of the maximum possible signal to the corrupting noise, indicating image reconstruction quality. SSIM, on the other hand, assesses changes in structural information, luminance, and contrast, offering a perception-based measure of image quality. These metrics together provide a balanced evaluation of both technical and perceptual aspects of RAW image reconstruction.

### 4.2 Working Flow

To improve understanding and implementation of the proposed method, we have included pseudo codes for the main modules in our approach. Algorithm. 1 demonstrates the contrastive training phase of our Pixel Searching Encoder, while Algorithm. 2 showcases the reference clustering and grouping algorithm. Lastly, Algorithm. 3 presents the Prior Metadata Based RAW Reconstructing Algorithm.

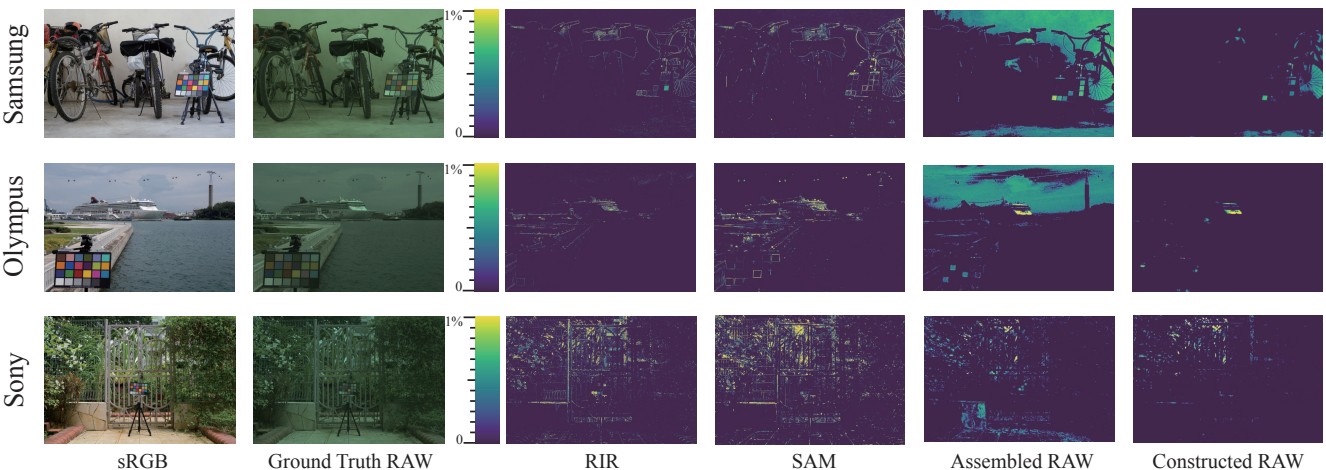

Figure 5: Illustration Comparison with Metadata-based Methods. As demonstrated in this figure, our method achieves the highest level of accuracy with the least amount of error compared to other methods when compared to the Ground Truth RAW images.

## 4.3 Ablation Study

**Clustering and Grouping** To assess the impact of the clustering and grouping algorithm, we conducted a series of ablation studies on Olympus E-PL6 [8]. As shown in Table 1, Pixel Searching directly assembles the output RAW by searching for similar RAW pixels from the entire reference dataset. Clustering and Grouping involve assembling the RAW output with search pixels based on prior metadata grouping. The results indicate that the clustering and grouping algorithm does not noticeably reduce the performance in terms of PSNR and SSIM, but it significantly reduces the computational burden of the searching procedure.

There are two hyperparameters that can affect the performance and efficiency: the number of clusters and the group size. The number of clusters determines the number of pixels included in the prior metadata. A larger number of clusters results in greater pixel variety, but it also increases the computational burden when searching for similar pixels. On the other hand, the group size determines the number of pixels used for grouping. A larger group size leads to more stable feature generation, but it reduces the diversity of the final features. As shown in the table, when we choose an appropriate cluster number and group size, the clustering and grouping algorithm does not noticeably compromise performance compared to searching from the entire reference dataset. In this case, we have selected a cluster size of 5,000 and a grouping size of 1,000, resulting in a final pixel number that is only 0.02% of the original reference dataset.

**Network Design** To evaluate the impact of different modules in the network, we conducted several ablation studies, as summarized in the table. Our baseline network is a U-Net network that takes only sRGB images as input. "Prior RAW" refers to the baseline network that takes both sRGB images and assembled RAW images as input. "Reliable Attention" represents the Prior RAW network with the reliable attention module to suppress unreliable areas. "Non-Local" indicates the Non-Local fusion module. As shown in the table, incorporating Prior RAW data can significantly improve

| Method | PSNR | SSIM |
|---|---|---|
| Pixel Searching | 44.32 | 0.9935 |
| Clustering and Grouping | 43.98 | 0.9932 |
| Cluster Number 10,000 | 43.85 | 0.9930 |
| Cluster Number 5,000 | 43.98 | 0.9932 |
| Cluster Number 3,000 | 43.95 | 0.9932 |
| Cluster Number 1,000 | 39.15 | 0.9896 |
| Group Size 1500 | 42.86 | 0.9929 |
| Group Size 1000 | 43.98 | 0.9932 |
| Group Size 500 | 41.56 | 0.9943 |
| sRGB+UNet | 44.32 | 0.9831 |
| + Prior RAW | 50.35 | 0.9970 |
| + Reliable Attention | 51.85 | 0.9974 |
| + Non Local | 52.05 | 0.9975 |

Table 1: Ablation studies of our method on the Olympus E-PL6. We compare the performance of different parts of our pipeline, such as pixel searching, reference compression, and the influence of different modules in the reconstructor.

the baseline model. The Reliable Attention module helps eliminate side effects from unreliable areas and improves the performance compared to the original Prior RAW input. Finally, by using the Non-Local fusion module, the model further refines the prediction and achieves the best performance.

## 4.4 Comparison

As shown in the ablation study, even the assembled RAW images achieve a high level of accuracy. When analyzing Fig. 5, it becomes evident that the errors in the assembled RAW images are not uniformly distributed. Instead, they exhibit high error values in specific areas such as the sky area, while other areas have low error values.

| Method | Extra Camera Computation | Extra Camera Storage | Samsung NX2000 | | Olympus E-PL6 | | Sony SLT-A57 | |
|---|---|---|---|---|---|---|---|---|
| | | | PSNR | SSIM | PSNR | SSIM | PSNR | SSIM |
| RIR [26] | $O(WH)$ | $O(WH)$ | 45.66 | 0.9939 | 48.42 | 0.9924 | 51.26 | 0.9982 |
| SAM [29] | $O(WH)$ | $O(WH)$ | 47.03 | 0.9962 | 49.35 | 0.9978 | 50.44 | 0.9982 |
| CAM [24] | $O(WH)$ | $O(WH)$ | 48.08 | 0.9968 | 50.71 | 0.9975 | 50.49 | 0.9973 |
| CAM + Finetune [24] | $O(WH)$ | $O(WH)$ | 49.57 | 0.9975 | 51.54 | 0.9980 | 52.55 | 0.9980 |
| Ours | 0 | 0 | 48.98 | 0.9970 | 52.05 | 0.9975 | 53.11 | 0.9983 |

**Table 2: Comparison with Metadata-based RAW Reconstruction Methods. The extra camera computation and storage are the additional resource costs on the camera devices, where $W$ and $H$ represent the width and height of the image, respectively.**

| Method | PSNR | SSIM |
|---|---|---|
| CA [31] | 34.74 | 0.9317 |
| ACDC [32] | 34.68 | 0.9152 |
| IPAD [19] | 34.91 | 0.9345 |
| BitNet [5] | 38.48 | 0.9657 |
| BE-CALF [18] | 38.94 | 0.9680 |
| Nam *et. al.* [24] | 39.57 | 0.9719 |
| Nam *et. al.* + Finetune [24] | 39.73 | 0.9721 |
| Ours | **39.79** | **0.9732** |

**Table 3: Quantitative comparison of bit-depth recovery (4-to-8-bit) methods on the Kodak dataset [1].**

This can be attributed to the high-illumination and low-texture nature of these areas, making it difficult to find accurate similar pixels in the reference set. To understand the underlying cause of this phenomenon, we examined the affinity matrix for both the areas with high error values and those with low error values. We found that the maximum values of the affinity matrix for the low-error areas are approximately 0.9, while for the high-error areas, they fall below 0.6. Based on this finding, we attribute this discrepancy to the lack of similar pixels in the reference dataset for the high-error areas, leading to the mismatching problem. Consequently, the Pixel Searching Encoder is unable to find suitable references for these pixels, resulting in the observed discrepancies.

We have named our final method PriorRAW, which incorporates the steps of prior metadata generation, metadata fusion, and the Encoder-Decoder network. As shown in Table 2, PriorRAW achieves state-of-the-art performance overall and significantly improves the performance of assembled RAW images. From the visualization, it is evident that the high error values in the assembled RAW images are greatly reduced. This indicates that the neural network has learned to address the error caused by the mismatched pixels problem and has achieved improved overall performance by combining information from sRGB inputs and the prior metadata. Compared to metadata-based methods, we have observed that errors in metadata-based methods are concentrated in high-frequency regions. This is because the sampled metadata is unable to accurately recover the original pixels in areas with rich details. However, our method does not have this issue. Instead, the error areas of our model are concentrated in highlights areas, which suffer from the most severe

information loss in the sRGB image and are therefore difficult to recover the original RAW pixels.

## 4.5 More Applications

The proposed method for RAW reconstruction based on prior metadata can also be utilized as a versatile image reconstruction pipeline. In order to assess its effectiveness in tasks beyond RAW reconstruction, we choose to evaluate our methods through a bit-depth recovery task. Bit-depth recovery involves reconstructing a high bit-depth image from a low bit-depth version, with the aim of restoring finer tonal details and reducing quantization artifacts. This process plays a crucial role in enhancing image quality by expanding its dynamic range and preserving subtle nuances that may be lost in lower bit-depth variations. To accomplish this, we have selected two widely-used datasets, namely MIT-Adobe 5K [4] and Sintel [3], for training the 4-to-8-bit recovery model. Then, we choose the representative Kodak [1] dataset for evaluation purposes.

As shown in Table 3, we compare our methods with several representative algorithms for bit-depth recovery. By incorporating the prior metadata extracted from high-bit depth data, our model with the fusion module and encoder-decoder structure significantly outperforms state-of-the-art bit-depth recovery algorithms, achieving a PSNR of 39.79.

## 5 CONCLUSION

In this paper, we focus on efficiently reconstructing RAW images using sRGB inputs. Unlike previous methods, we propose a new reconstruction pipeline based on prior metadata. Our approach is based on the idea that pixels captured under similar lighting conditions for similar objects can provide valuable prior information for reconstructing the current pixel. To achieve this, we design a contrastive learning method to train the Pixel Searching Encoder, enabling it to efficiently identify the most similar pixels in the reference dataset. We also develop a clustering and grouping algorithm to compress the reference features and sRGB-RAW pixel pairs into a smaller scale during the inference phase, while maintaining performance. Finally, using the assembled prior metadata and the sRGB input, we design a fusion module and an Encoder-Decoder network to reconstruct the final RAW images. Experiments show that our method achieves excellent performance on the RAW reconstruction task and can be used for similar tasks like bit-depth recovery. Our goal is to contribute to a wider range of reconstruction tasks, enhancing their applicability.

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
