# OpenReview forum: "Prior Metadata-Driven RAW Reconstruction: Eliminating the Need for Per-Image Metadata"
_acmmm.org/ACMMM/2024/Conference — MM2024 Poster_

### Official Review · Reviewer_1WBT · 2024-05-13

**Rating:** 3
**Confidence:** 2

**Summary:**

The purpose of this work is to reconstruct raw images from sRGB data in a more efficient way. Specifically, this work utilizes Prior Meta as a reference to reconstruct the RAW images, rather than using per-image metadata directly. To achieve this goal, it designs a pixel searching network to find the most similar pixels in the reference RAW images and compresses the reference images. Afterwards, it employs reconstructor to reconstruct RAW images.

**Strengths:**

1. This work uses Prior Meta as a reference to reconstruct the RAW images, rather than using per-image metadata directly, which is more efficient than previous methods.
2. This work proposes a pixel searching network to search for the most similar RAW pixels in the reference dataset and designs a clustering and grouping algorithm to identify the most representative pixels.
3. This work proposes a raw reconstructor with a prior metadata fusion module to achieve better performance.

**Limitations:**

1. It might be better to give more experimental results compared with previous work, especially visual comparison results.
2. Since the motivation of this work is to provide a more efficient reconstruction method, it may be more convincing to provide more comparisons of the cost of model training and inference with previous work, especially the amount of computation, time, etc.
3. Does the network model need to be quantified when deployed in practice? If quantization is required, how much does the performance drop?

**Suitability:**

3

---

### Official Review · Reviewer_DuxP · 2024-05-24

**Rating:** 4
**Confidence:** 3

**Summary:**

The paper proposes to a method to reconstruct the raw image from sRGB image. By using Prior Meta as a reference to reconstruct the RAW image, the proposed method achieves slightly better perforance than some of the existing metadata-based models.

**Strengths:**

- The topic of raw image reconstruction from sRGB is interesting.
- It seems to have better performance than some metadata-based raw image reconstruction methods.

**Limitations:**

- What’s the computational cost of training/inference? How about number of parameters, FLOPS?
- The key challenge of the raw image reconstruction cannot be solved in this work, making the proposed setting less practical. A key challenge is that the process from raw image to sRGB is a lossy process and there exists inevitable serious information loss in low light and highlight areas. However, without using metadata cannot recover the lost details.
- How about the performance of the model compared with SOTA metadata method, e.g., [a]. Besides, SAM et.al. utilize 16bit sRGB image as input, which is less practical. How about the performance using 8-bit sRGB/8-bit JPEG images?
- It is a little confusing why w/ and w/o Prior Metadata can have such a huge improvement, e.g., 6db according to the ablation study. The prior metadata usually can be learned into neural networks and using a codebook usually cannot obtain such a huge improvement.

[a] https://github.com/wyf0912/R2LCM

**Suitability:**

2

---

### Official Review · Reviewer_n538 · 2024-05-25

**Rating:** 5
**Confidence:** 4

**Summary:**

This paper focuses on transferring sRGB images back to their corresponding RAW images with reference to existing RAW images, which is different from previous studies that solely use sRGB images or require additional sampling procedures in the camera devices. The proposed method contains three stages: pixel search encoder, prior information compression, and prior MetaData-based reconstructor. The pixel search encoder aims to learn an effective correlation matrix between the target and reference images. To improve the computational efficiency, it uses the K-Means algorithm to extract representatives prior to serving for RAW image reconstruction. In the third stage, it reconstructs the RAW images based on the pre-trained pixel search encoder and prior metadata. The paper demonstrates the effectiveness of the proposed method on the NUS dataset which achieves promising performance.

**Strengths:**

1. This paper clearly discusses the limitations of the existing methods and the structure of this paper is easy to follow.

2. The paper comprehensively evaluates the performance of the proposed method using different numbers of clusters and shows that it significantly affects the performance.

3. The visual comparison is straightforward and convincing.

**Limitations:**

1. This method is a reference-based method, which requires that the reference images should be similar to the target images. How to choose the reference images for each target sRGB image?

2. It is hard to understand why the pixel-searching encoder uses contrastive learning. It seems that the raw version of the target sRGB is used to calculate the l2 loss. And, why does this method automatically learn the similar and dissimilar pixels?

3. The paper shows that the number of clusters significantly affects the performance. How to select these hyper-parameters for general sRGB images? Or are these parameters specifically set for each sRGB image?

**Suitability:**

3

---

### Meta-Review · Area_Chair_9hGW · 2024-07-04

**Recommendation:** Accept (Poster)
**Confidence:** 5

**Metareview:**

The response from the authors did not change the initial ratings: Weak Accept, Borderline Accept, Borderline Reject.

After carefully reading the paper, the reviews and the response the Meta Reviewer agrees with them that the paper's topic is of interest, the solution is novel and achieves promising performance, and that the paper covers well limitations.
At the same time, "the proposed method might be sensitive to the selection of hyperparameters and the reference dataset" (Reviewer n538) and the authors' response provides information that should be in the paper.

The strengths outweigh the weaknesses and the Meta Reviewer agrees that the paper's contributions are worthy publication and invites the authors to add "discussions regarding the experimental settings and new results"(Reviewer DuxP) to the camera ready.